# Elongated Gravity Sources as an Analytical Limit for Flat Galaxy Rotation Curves

Felipe J. Llanes-Estrada 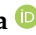

Departamento de Física Teórica and IPARCOS, Institute for Particle and Cosmological Physics, Universidad Complutense de Madrid, 28040 Madrid, Spain; fllanes@fis.ucm.es; Tel.: +34-913-944-460

**Abstract:** The flattening of spiral-galaxy rotation curves is unnatural in view of the expectations from Kepler's third law and a central mass. It is interesting, however, that the radius-independence velocity is what one expects in one less dimension. In our three-dimensional space, the rotation curve is natural if, outside the galaxy's center, the gravitational potential corresponds to that of a very prolate ellipsoid, filament, string, or otherwise cylindrical structure perpendicular to the galactic plane. While there is observational evidence (and numerical simulations) for filamentary structure at large scales, this has not been discussed at scales commensurable with galactic sizes. If, nevertheless, the hypothesis is tentatively adopted, the scaling exponent of the baryonic Tully–Fisher relation due to accretion of visible matter by the halo comes out to reasonably be 4. At a minimum, this analytical limit would suggest that simulations yielding prolate haloes would provide a better overall fit to small-scale galaxy data.

**Keywords:** galactic rotation; nonspherical gravitational sources; modified gravity

## 1. Introduction

With decades of effort [1], it has been established that the rotation speed of spiral galaxies is largely independent of the distance to their center, $v \sim$ constant, even well beyond the end of the luminous matter distribution, whereas Kepler's third law applied to a point-like mass or spherical source yields $v \sim 1/\sqrt{r}$. This unexpected result (see Figure 1 for a small sample of the SPARC data) is usually interpreted by (a) there being non-luminous matter spherically distributed with a very specific radial dependence, which is actively searched for in the laboratory (Dark Matter), or (b) Newton's law of gravitation is failing for small centripetal acceleration (Modified Newtonian Dynamics), and the force law is different, yielding to a modified Kepler's first law.

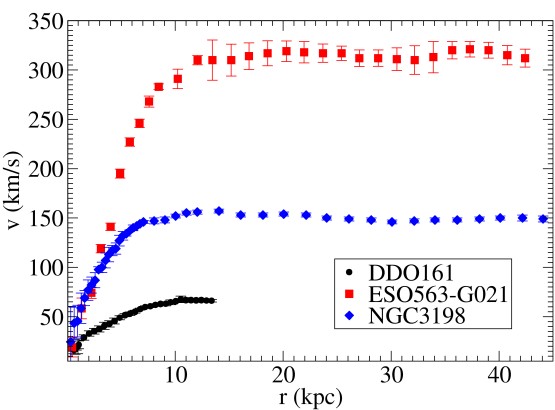

**Figure 1.** Galactic rotation curves show a flat (distance-independent) rotation velocity at large distances to the galactic center (data sample from Reference [2], SPARC collaboration).

Investigations on the first possibility have concentrated on spherical galactic haloes. However, much evidence of dark matter in present-day cosmology seems consistent with it having filamentary structure at large scales. Thus, it is reasonable to ask oneself down to what scale is that filamentary organization meaningful.

At least for scales commensurable with those galaxy-sized ones, statistical gravitational lensing analysis (stacking galaxy pairs) [3] is an interesting indication that there actually are matter filaments extending between galaxies, though no actual filament has been individually resolved, to my knowledge.

Figure 2 sketches the point of this article that there may be merit in allowing dark matter at the galactic scale (~10–100 kpc) to be organized in a cylindrical or otherwise elongated, rather than spherical, geometry. A difference between spherical and elongated gravitational-source distributions is the local density of dark matter at a given point in the galactic equator, where elongated sources would assign a smaller density, concentrating it instead along the polar regions.

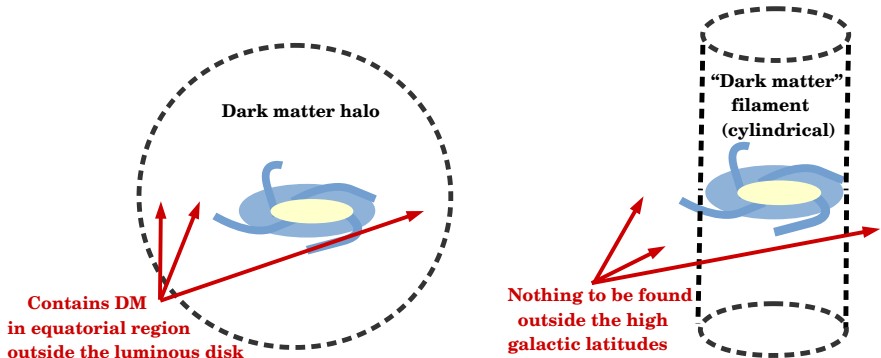

**Figure 2.** (**Left**): A spherical halo of dark matter extending much beyond the disk of spiral galaxies. The radial dependence of that mass distribution that yields a flat rotation curve is the quite unique $\rho_{\text{DM}} \propto r^{-2}$. This can be obtained from hydrostatic equilibrium with an isothermal distribution, but requires that dark matter is thermalized, involving some heat transfer mechanism (at odds with dark matter particles being very weakly interacting). (**Right**): A cylindrical distribution extending from the galactic polar regions explains flat rotation curves without any fine tuning: they are the natural consequence of such gravitational source independently of its nature (whether dark matter or otherwise) without having to abandon Newtonian mechanics for MOND or other modifications. An immediate consequence of such geometry is that dark matter searches (by lensing or otherwise) at low galactic latitudes have less scope.

## 2. Most Common Explanations for Flat Rotation Curves
### 2.1. Kepler's Problem

The solar system responds mostly to the concentration of mass at its center in the sun. Circular orbit equilibrium, together with Newton's gravitation law, demands that

$$m\frac{v^2}{r} = \frac{GM_\odot m}{r^2}, \tag{1}$$

that is the simplest illustration of Kepler's third law ($T^2 \propto r^3$). Solving for the rotation velocity $v$, one finds a velocity falling with the square root of the distance to the center

$$v = \frac{\sqrt{GM}}{\sqrt{r}} . \tag{2}$$

This simple law works flawlessly in the solar system down to a precision of $10^{-3}$–$10^{-4}$, at which level perturbations, largely due to Jupiter and other planets, become important (see Section 5.2 below).

However, extrapolating the law to galactic rotation curves, as seen in Figure 1, becomes a startling failure, which has been a driver of much research in astrophysics: rotation curves reach a plateau with an approximately constant velocity $v_\infty \sim 30$–$300$ km/s for tens of kiloparsec, very much unlike Equation (2). Sometimes, modifications of General Relativity are invoked as an explanation of the discrepancy, but a nonrelativistic, Newtonian treatment should suffice to a reasonable precision since

$$\frac{v_{\text{galaxy}}}{c} = \frac{(30 - 300)\ \text{km/s}}{(3 \times 10^5\ \text{km/s})} = O(10^{-3}) \tag{3}$$

or less. A much explored possibility is to modify Newton's laws, as recalled later in Section 2.3.

Alternatively, since luminous matter in galaxies stops being dense enough well before that flatness sets in, a dark matter halo that produces no light is most often postulated, that I briefly recall in the next section, Section 2.2.

Indeed, if the orbit is inside the mass distribution, the mass in Equation (2) refers to that of the inner sphere with the orbit as equator (from Gauss's law), $M(r)$. For example, a constant density cloud would yield $M(r) = \frac{4\pi}{3} r^3 \rho$ and a linearly-growing velocity field

$$v = \sqrt{\frac{4}{3}\pi G \rho r}, \tag{4}$$

not unlike what is observed for small $r$ in usual rotation curves, such as the example in Figure 1.

To obtain a constant velocity $v \simeq v_\infty$, independent of $r$, the density in the square root of Equation (4) needs to cancel the $r$ outside and, therefore, behave as $\rho \propto \frac{1}{r^2}$.

### 2.2. Standard Isothermal, Spherical Dark Matter Halo

Hydrostatic, nonrelativistic, equilibrium in a standard spherical halo made of fluid-like matter dictates

$$\frac{dP}{dr} = -\frac{G\rho(r)}{r^2} \int_0^r 4\pi r'^2 \rho(r') dr', \tag{5}$$

so that inner layers of the halo, at a higher pressure, can support the weight of the outer ones.

The ideal gas law is usually employed to eliminate the pressure, so that

$$P(r) = \frac{k_B}{m} T(r)\rho(r), \tag{6}$$

with $m$ the typical "particle" matter, and $T(r)$ the temperature field.

Taking a derivative of Equation (5) with respect to $r$, to convert it into a pure differential equation, one finds [4]

$$\frac{k_B}{m}\frac{d}{dr}\left(\frac{r^2}{\rho(r)}\frac{d}{dr}(T(r)\rho(r))\right) = -4\pi G r^2 \rho(r). \tag{7}$$

Under the hypothesis that the dark matter halo has reached a uniform temperature (and it is not known how this happens, as it depends on the dark matter interactions, but requires some sort of heat conduction between different spherical shells), one finds easily that the equation admits the power-law solution

$$\rho \propto \frac{1}{r^2}. \tag{8}$$

This behavior is exactly what is needed to produce the observed flat rotation curves.

On the contrary, experimental searches for particulate dark matter at colliders [5] (through production of dark matter particles), at underground laboratories [6,7] (through

direct detection by collisions with nuclei), or at gamma-ray or other particle observatories [8] (through indirect detection of presumed decay products) have all come up empty-handed.

Searches for macroscopic-sized dark matter constituents, such as Massive Compact Halo Objects, by gravitational lensing and by binary star disruption [9], have yielded stringent constraints that leave little space for the existence of large dark matter chunks in the halo. An exception that is still standing is the possibility of $O(100M_\odot)$ black holes [10].

### 2.3. Modified Newtonian Dynamics

An alternative to dark matter that naturally explains galaxy rotation curves is Modified Newtonian Dynamics (MOND) [11]. The basic idea is to postulate a new scale $a_0 \sim 1.2 \times 10^{-10}$ m/s$^2$ such that the acceleration caused by a force depends on its size respective to this scale,

$$
\begin{aligned}
a > a_0 \quad & a_{\text{Newton}} = \frac{MG}{r^2} \\
a < a_0 \quad & a = \sqrt{a_0 a_{\text{Newton}}}
\end{aligned}
\tag{9}
$$

This recipe was thereafter formulated as an Effective Field Theory in the gravitational potential and given shape as a bimetric theory of gravity [12].

A key prediction of MOND that is observationally a reasonable success is that the exponent of the baryonic Tully–Fisher relation (discussed below in Section 5) is exactly equal to 4, so that $M_{\text{luminous}} \propto v_\infty^4$. The intensity of the second peak of the cosmic microwave background was also successfully predicted.

Another modification of the theory of gravity has been applied to the explanation of galactic data by Varieschi in a series of articles contemporary to the present manuscript [13–15]. The idea is that of so-called "Newtonian Fractional-Dimension Gravity" (NFDG). If the space dimension is generalized from the integer 3 to a real number $D$, while demanding Gauss's law, the Newtonian potential becomes

$$
\begin{aligned}
\tilde{\phi}(\mathbf{r}/\mathbf{l_0}) &= -\frac{2\pi^{1-D/2}\Gamma(D/2)}{l_0(D-2)} \int_{\Omega_D} d^D(\mathbf{r'}/l_0) \frac{\rho(\mathbf{r'})l_0^3}{|\mathbf{r}/l_0 - \mathbf{r'}/l_0|^{D/2}} \quad D \neq 2 \\
\tilde{\phi}(\mathbf{r}/\mathbf{l_0}) &= \int_{\Omega_2} d^2(\mathbf{r'}/l_0) \frac{2G}{l_0}\rho(\mathbf{r'})l_0^3 \ln|\mathbf{r}/l_0 - \mathbf{r'}/l_0| \quad D = 2 \,.
\end{aligned}
\tag{10}
$$

For $D = 2$, this modified-gravity theory achieves the asymptotic flatness that is thoroughly compared with MOND predictions in those works (additionally, MOND can be directly derived from fractional gravity [16,17]). Moreover, it can also reproduce a non-flat behavior of the galactic rotation curves by being able to smoothly interpolating between $D = 3$ (conventional gravity) and $D = 2$. In this aspect, the approach is akin to those with dark matter distributions controlled by one parameter. It will be interesting to see what its implications are for yet larger scales: if it stabilizes in $D = 2$ or it continues decreasing into distances relevant for cosmology. Substituting the factor 3 in the Friedmann-Robertson-Walker equations by $D$ must lead to a very different cosmological model.

Problems with large scale data are already an issue for MOND: the intensity of the 3rd peak in the cosmic microwave background, the galaxy power spectrum, the gravitational lensing in galactic cumuli, and, generically, observables at scales much larger than the 100 kpc one for which MOND was conceived, have turned out to be disappointing [18]. To bring MOND into broad agreement with data, various authors typically supplement it with additional matter [11] (such as sterile neutrinos, vector fields as in the TeVeS formulation, etc.). Its advantages over dark matter-based formulations are then blurred.

Even more so, the claimed evidence for galaxies without dark matter [19] and the analysis of the bullet cluster [20], where dark and conventional matter would be separated, plays against MOND as therein no effect beyond what luminous matter suggests is visible: whereas dark matter is contingent on accretion, modified gravity cannot be taken away from conventional matter.

Finally, adoption of MOND would force us to abandon the theoretically compelling nonrelativistic Newtonian theory, in which the flux of the gravitational field is conserved,



so that its spreading in our three-dimensional world dilutes it over a $4\pi r^2$ sphere yielding the $1/r^2$ law of gravity, and translational invariance implies Newton's second law with the conventional acceleration. MOND is unconvincing for most theorists.

However, the working recipe of Equation (9) to obtain the right galaxy rotation curves can be obtained in a different, simple way.

### 3. Cylindrical Symmetry of the Gravitational Source

What MOND achieves by taking the square root of a constant times the $1/r^2$ force is to moderate its falloff to $1/r$, the geometric mean.

However, Gauss's law suggests that $F \propto 1/r$ is the natural one in a two-dimensional world. This is achieved routinely in electrostatics with a cylindrical/filamentary source running from $-\infty$ to $\infty$, yielding translational invariance along the $OZ$ axis and effectively leaving a two-dimensional theory in the perpendicular directions. Thus, while, in three-dimension, $F \propto 1/r^2$ implies $v^2 \propto 1/r$ (Kepler's law), in two-dimension, $F \propto 1/r$ brings the desired $v^2 \propto$ constant about.

To see it, recall that the gravitational acceleration around a cylindrical mass distribution, in cylindrical coordinates $(r, \phi, z)$, takes the form $\vec{g} = g(r)\hat{r}$ by symmetry. Its flux outwards of a pillbox of height $a$ surrounding the cylinder is

$$\Phi = \int_0^a dz \int_0^{2\pi} r d\phi \hat{r} \cdot \vec{g} = (2\pi a) r g(r) \tag{11}$$

through its side, and zero through its lids. Because of Gauss's law, the flux is also

$$\Phi = -2\pi a \int_0^r \rho \Delta V(\rho) d\rho = -(4\pi G)m \tag{12}$$

in terms of the gravitational potential, with $\vec{g} = -\nabla V$ and with $m$ the mass contained by the pillbox. If the linear mass density of the cylinder is

$$\lambda = \frac{dm}{dz}, \tag{13}$$

combining Equations (11) and (12) yields

$$\vec{g} = \frac{-2G\lambda \hat{r}}{r}, \tag{14}$$

that, of course, stems from a potential

$$V(r) = 2G\lambda \ln\left(\frac{r}{r_0}\right). \tag{15}$$

As it diverges at large $r$, its zero needs to be arbitrarily chosen at a certain $r_0$.

This is a staple discussion of any textbook covering electrostatics, and the potential is the natural one conserving the integrated flux out of the source in a reduced 2-dimensional problem. However, it is not often discussed in the context of gravitational interactions because of the scarcity of known large cylindrical sources of gravity.

The extraction of the rotation velocity mirroring that of Section 2.1 proceeds by again equating the gravitational and centripetal forces for a circular orbit

$$mg = \frac{2mG\lambda}{r}. \tag{16}$$

This indeed yields a distance-independent rotation velocity,

$$v^2 = 2G\lambda. \tag{17}$$

This formula contains one independent parameter, $\lambda = m/L$, the linear mass–density of the cylindrical source, and notably absent is $R$, the radius of the cylinder in the transverse direction to be discussed shortly.

Let me put some figures to Equation (17). Conveniently, take the velocity in terms of a dimensionless parameter of order 1, $v = v_{100}$ (100 km/s), and substitute Cavendish's constant $G$ so that

$$\lambda = 1.16 v_{100}^2 \times 10^{12} M_\odot / \text{Mpc} \,. \tag{18}$$

For the Andromeda galaxy, $v_{100} \simeq 2.2$; therefore, $\lambda = 5.6 \times 10^{12} M_\odot /$ Mpc. To understand this number, we need to think that a Megaparsec of such filament (the typical galaxy–galaxy separation) contains about 7 times the stellar mass of Andromeda, $M^{\text{stellar}} \simeq 8 \times 10^{11} M_\odot$. This means that such filamentary structure is compatible with the overall dark matter fraction needed for the cosmic sum rule $\Omega_\Lambda + \Omega_{DM} + \Omega_b = 1$ in present day's cosmology, with $\Omega_b \sim 4\%$ and $\Omega_{DM} \sim 23\%$.

### 3.1. Corrections to a Basic Filamentary Geometry

Next, let us relax the assumption of an infinitely long cylinder of unspecified radius and see the corrections brought about by various geometrical modifications.

#### 3.1.1. Finite-Length Cylinder

First, consider a finite cylinder that, instead of extending over the entire OZ axis $(-\infty, \infty)$, does so only over the interval $(-a, a)$.

Matter naturally accretes to the horizontal plane by the center of the cylinder where the gravitational potential is minimum, and, there, Equation (17) is replaced by

$$v_{\text{equator}} \simeq \sqrt{2G\lambda \left(1 - \frac{r^2}{2a^2}\right)} \,. \tag{19}$$

Thus, the end of the filament threading, the galaxy is reflected in the velocity field starting to falloff at sufficient distance.

There is scarce data suggesting such fall for most galaxies in the SPARC catalogue. However, it can be accommodated within the uncertainty bands of the velocity measured. With typical error of 10 km/s in $v = 200$ km/s, we can propagate the error backwards and find

$$a \geq 1.6 \, r_{\text{break}}, \tag{20}$$

with $r_{\text{break}}$ being the point at which the constant velocity law might start breaking down.

Data for our neighboring Andromeda galaxy has been reported, suggesting that, at 100 kpc, the velocity field starts diminishing in modulus [21]. Taking that data at face value, for Andromeda $r_{\text{break}} \sim 0.1$ Mpc) implies an elongated source out to at most a similar length.

#### 3.1.2. Finite-Width Cylinder

If the density of a finite-sized cylinder of radius $R$ is uniform, $\rho = $ constant, the linear density becomes quadratic $\lambda(r) = 2\pi \int_0^r r' dr' \rho(r') \propto r^2$, and the velocity field inside the cylinder shows a linear rise,

$$v = \sqrt{2G\lambda} \frac{r}{R} \,. \tag{21}$$

Matching this to the flat rotation curve of Equation (17) yields a reasonable first-order explanation of typical spiral rotation, as shown in Figure 3.

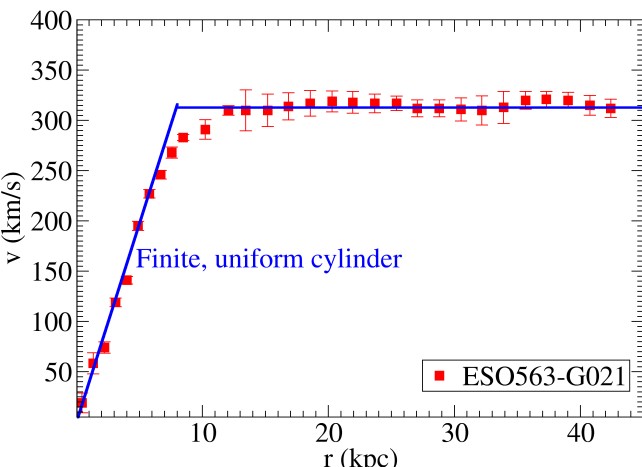

**Figure 3.** A linear rise out to a radius $R$ given by Equation (21), followed by a flat $v_\infty = \sqrt{2G\lambda}$, is a reasonable first approximation to numerous spiral galaxy rotation curves, and follows from a finite-sized cylinder of uniform density. Of course, edge effects or other density profiles could be included in extensive studies.

With a power law density profile, one would obtain an additional factor of $(1 - \frac{2}{\alpha+2}(r/R)^\alpha)$ that I have not yet attempted to match to the data (An extended study of the entire SPARC database with several different geometries is underway [22], and the resulting $\chi^2$ distributions will be reported elsewhere.).

### 3.1.3. Rotation Curve Outside a Sphere + Filament Distribution Arrangement

One can combine the leading cylindrical mass distribution with a smaller spherical one that can represent the visible matter bulge (and only very crudely, the monopole contribution of the galactic disk).

In terms of the cylinder's linear mass density and of the sphere's mass, the velocity field takes the form

$$v = \sqrt{2G\lambda + \frac{GM}{r}}, \tag{22}$$

that converges $v \to v_\infty = \sqrt{2G\lambda}$ at large radii. The SPARC data file offers examples of rotation curves where this effect might be visible, and one of them is plotted in Figure 4.

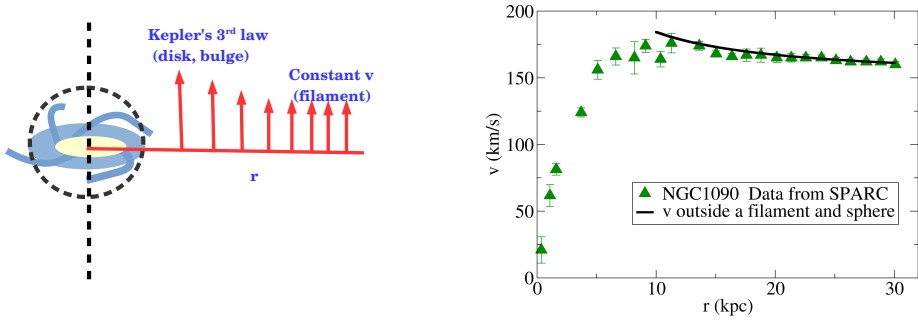

**Figure 4. Left**: Scheme showing how the velocity falls towards the asymptotic limit given by the filament alone, as the contribution of the sphere becomes smaller as per Kepler's third law. **Right**: An example galaxy curve extracted off the SPARC database where the effect might be visible.

## 4. Classical Equations of Motion Outside the Filament

The cylindrical symmetry and the conservative gravitational potential provide us with two constants of motion: the energy $E_0$ and angular momentum along the cylinder's axis $L_z$. This means that, out of the three second-order differential equations for the cylindrical coordinates $(r, \varphi, z)$, two can be integrated once. If these are chosen to be that for $\varphi$ and that for $r$, Newton's equations become

$$\dot{\varphi} = \frac{L_z}{mr^2} \tag{23}$$

$$\ddot{z} = \frac{-GMz}{(z^2 + r^2)^{3/2}}$$

$$\dot{r} = \sqrt{\frac{2E_0}{m} - \frac{L_z^2}{2m^2r^2} - \dot{z}^2 + \frac{2GM}{\sqrt{r^2 + z^2}} - 4G\lambda \ln(r)},$$

in which the potential due to the cylinder appears in the square root,

$$V(r) = 4G\lambda \ln(r). \tag{24}$$

The first two of Equation (23) are the same as in the planar Kepler problem. It is the third one that shows a difference due to the presence of the cylinder. We next explore a few consequences of these equations.

### 4.1. Helicoidal Motion along the Filament

The first kind of motion is helicoidal along a filament, sketched in Figure 5, with $r$ constant, $\dot{r} = 0$, $\dot{z}$, and $\dot{\varphi}$ also constant.

Though, in late-time cosmology, the voids around the filaments have little matter to accrete, matter can still hop from galaxy to galaxy along the cylinders in a helicoidal manner. This costs only the gravitational energy needed to escape the field of the luminous matter in the galaxy disk, while the displacement along the filament, because of Equation (23), takes place with constant $v_z$, or $\frac{dv_z}{dt} = 0$ far from the galaxy.

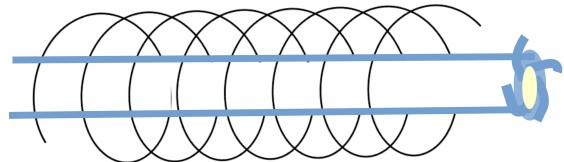

**Figure 5.** At large distance from any galaxy, movement along cosmic web filaments is helicoidal. This is well known from the movement along a $B$ field in electrodynamics; the difference here is that not only charged particles, but also neutral objects (whether gas or any sort of aggregate), by the equivalence principle, follow the helix. The trajectory is stretched by the galaxy upon approaching it when the acceleration due to the extra mass at the end is felt.

#### 4.1.1. Cyclotron/Synchrotron Radiation

When charged particles follow such helicoidal trajectory, they radiate. The phenomenon is analogous to circular trajectories in a magnetic field, but there are several curious differences that I now expose.

Unlike in conventional magnetic synchrotrons, the rotation around the filament, by construction (after all, that is what yields flat galaxy rotation curves) occurs with constant perpendicular velocity

$$v_\perp^2 = \frac{2G\lambda}{\gamma}, \tag{25}$$

where, if the motion is relativistic, the Lorentz time-dilation factor $\gamma > 1$ has to be taken into account. The gyration period then grows with distance, so that the angular frequency falls with $r$,

$$\omega = \frac{\sqrt{2G\lambda}}{\sqrt{\gamma} r} \tag{26}$$

(compare this with the equivalent $(eB)/(\gamma m_e c)$, an $r$-independent constant, in a $B$ field).

The characteristic frequency can then be read off the conventional synchrotron theory but substituting for the $\omega$ in Equation (26) as

$$\nu_{\text{char}} = 0.29 \frac{3}{2} \gamma^{2.5} \frac{\sqrt{2G\lambda}}{r}, \tag{27}$$

and the total power emitted as

$$P = \frac{2}{3} \frac{e^2}{c^3} \gamma^4 \left( \frac{2G\lambda}{\gamma r} \right)^2. \tag{28}$$

A first remarkable property of these expressions is the dependence in $e^2\lambda$ as opposed to the $e^4$ dependence of conventional radiation (the $e^2$ from emitting photons is common to both, but the force to turn the trajectory is $\propto e^2$ in the Lorentz force case, $\propto \lambda$ in the gravitational one). A second difference is the inverse-$r$ dependence that distinguishes it from radiation in a uniform $B$-field. The integrated emission is dominated by the inner electrons in the radial distribution, since $\int_R n(r) dr / r^2 \simeq n/R$ diverges for small $R$.

These observables serve to distinguish between an infinitely thin filament (such as a cosmic string) and a cylinder of more or less uniform density. In the first case, electrons near the filament ($R \to 0$) strongly radiate synchrotron power, which should be observable. In that case, additionally, a clear prediction is that this polar synchrotron radiation is correlated with the galactic rotation parameter $\sqrt{2G\lambda}$.

In the second case, a finite cylinder with $\lambda_{\text{inside}}(r) \propto r^2$, the maximum emission happens at the filament's edge; however, $R$ being now of galactic $\sim$ 10 kpc scale, the frequency and power are very suppressed, and the radiation is negligible.

It is known that numerous elliptical galaxies, for example, M87, radiate synchrotron-like along a filament (likely due to jet emission). There seem to be only four known spiral galaxies that radiate in the same way: J1352+3126, J1159+5820, J1649+2635, and J083+0532. These sources are not found in the SPARC database, so that I cannot presently answer whether their synchrotron radiation is or not correlated with their rotation velocity. We really should not expect this, but that would be a fun observation.

### 4.1.2. Galactic Aurora

It is well known that matter being pushed in jets out of an active galactic nucleus, for example, can heat up the medium and radiate. However, the situation is symmetric, and there can be matter propagating along the filament that falls into a galaxy from the poles. The produced radiation would be totally analogous to the phenomenon of the Aurora when charged particles fall on Earth following its magnetic field. Except, once more, neutral gas and matter chunks moving along the filament can now also cause it.

The effect could be similar to known axial structures, for example, the $\gamma$-ray bubbles of Fermi/LAT (respectively, the X-ray emission of ROSAT) over the north and under the south poles of the Milky Way [23].

### 4.2. Radial Motion towards the Filament

The purely radial motion towards the cylinder, far from the galaxy (large $z$) and with vanishing angular momentum $L_z = 0$, is interesting by itself, to evaluate the time that it takes for a filament to empty its environment by accreting all available matter.

The resulting one-dimensional radial equation from Equation (23) yields a time for moving a parcel of matter from distance $d$ into the cylinder radius $R$ given by

$$t = \int_d^R \frac{dr}{\sqrt{4G\lambda}\log^{1/2}\left(\frac{d}{r}\right)} \ . \tag{29}$$

Upon substituting a typical value for $\sqrt{4G\lambda} \simeq 10^{-3}c \simeq 0.307\,\mathrm{Mpc/Myr}$, one obtains

$$t = 3.26\mathrm{Gyr} \times \int_d^R \frac{dr(\mathrm{Mpc})}{\log^{1/2}(d/r)}. \tag{30}$$

To clean up a distance out to $d = 1\,\mathrm{Mpc}$, and bring the material into $R = 0.1\,\mathrm{Mpc}$, the value of the integral (numerically evaluated) is 1.7, yielding $t = 5.7\,\mathrm{Gyr}$. This is, of course, the standard formation of empty bubbles seen in SDSS, as well as computer simulations. All matter has had time to accrete to nearby filaments during a Hubble time.

### 4.3. Precession of Orbits Outside but near the Galactic Plane

Motion in the disk of the galaxy, supposed nearly perpendicular to the dark filament, is planar. However, perturbations above or below this plane put satellites in a precessing motion whose instantaneous plane is rotating around the axis (see Figure 6).

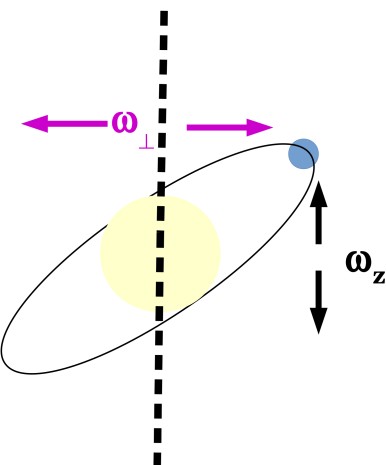

**Figure 6.** The generic motion of a satellite bound to the galaxy but off its disk's plane is not planar: the instantaneous orbital plane precesses around the cylinder. It can be seen by noting that the angular frequency of vertical motion $\omega_z$ coincides with that of the Newtonian problem, consistently with the translational invariance of the filament, but the oscillation angular frequency $\omega_\perp$ in the orbital plane differs due to the part of the force caused by $G\lambda$.

This comes about because vertical oscillation are unaffected by the cylinder and remain as in the two-body Keplerian problem: around $z = 0$,

$$\omega_z = \sqrt{\frac{GM}{r^3}} \tag{31}$$

(Kepler's third law), as can be seen from the second of Equation (23) corresponding to vertical motion.

Horizontal motion, however, returns to the same position with an angular frequency

$$
\begin{aligned}
\omega_\perp &= \sqrt{\omega_z^2 + \frac{2G\lambda}{r^2}} \\
&= \sqrt{\omega_z^2 + \frac{v_{\perp\infty}^2}{r^2}} \; .
\end{aligned}
\tag{32}
$$

This is modified by the presence of the cylinder.

At certain radial distances, the resonance condition (that yields closed orbits) is met, and the ratio $\omega_z/\omega_\perp$ is a rational number, a quotient of two integers. These $r$s may be written as

$$
r = \frac{GM}{v_{\perp\infty}^2}\left(\left(\frac{n}{m}\right)^2 - 1\right).
\tag{33}
$$

Such distances would yield less chaotic, less collisional orbits around the cylinder+galaxy mass distribution. For integer $n/m$, the orbital times are too large [24]: with the Andromeda numbers, $GM/v_\infty^2 \simeq 166$ kpc. The smallest integer $n^2 - 1 = 3$ then yields about half a Mpc, which amounts to a 10 M lightyear circumference. Because $v_\perp \simeq 10^{-3}c$, the orbital time is of order $1/H$, commensurable with the entire lifetime of the universe. However, for rational (smaller) $n/m$, perhaps some regularity in the satellite distribution can be found which correlates with the galactic rotation velocity, as in Equation (33).

In truth, it is not strictly necessary to require dark matter structures perpendicular to the galactic plane; if the angle between the disk and the cylinder- or cigar-like distribution was notably less than 90 degrees, the movement on the plane of the disk itself would be similar, except for a projection factor between the radius $r$ on the disk and $r_\perp$, its projection over the plane perpendicular to the filament, now different enough to require different variables. This projection factor being constant, it would not be directly measurable in the disk motion but shifted to the linear mass density in Equation (17) through the velocity,

$$
\lambda = \frac{v_\perp^2}{2G} \to \lambda_{\text{apparent}} = \frac{v^2}{2G} = \frac{v_\perp^2}{2G\cos^2\theta} \; .
\tag{34}
$$

However, what would be observable is the long-term precession of the galactic plane that might give rise to interesting structures in stellar streams [25] or satellites.

## 5. The Tully–Fisher Relation

The renowned empirical Tully–Fisher relation [26] relates magnitude (light) and linewidth (velocity dispersion, a proxy for mass) was originally applied to assist with the cosmic distance ladder.

However, the striking feature is that the first depends on visible, ordinary, baryonic matter, and the second, however, on the total matter, whether dark or not. This strongly suggests that we should set the amount of dark matter to be proportional to that of luminous matter, a feature that, from the point of view of dark matter theories, is a complex dynamical effect without a very clear explanation. MOND, however, does predict this effect, as it modifies the effect of matter respecting the proportionality $a \propto M$, and it is the distance-dependence that is modified.

This relation between was later extended to the "Baryonic Tully–Fisher relation" closer to the discussion of this article, a power-law constraint between the rotation velocity $v$ and the luminous mass $M_L \propto v^\alpha$.

### 5.1. Luminous Matter Accretion on a Filamentary Overdensity

First, let us assume that the filaments or cylinders are simply overdensities of dark matter. Then, if ordinary matter can fall towards any two nearby ones, it will do so towards the one exerting the largest force (see Figure 7).

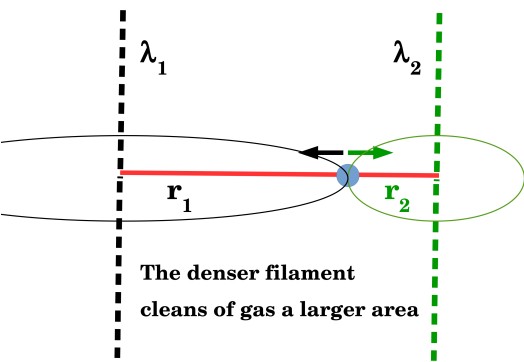

**Figure 7.** A denser dark matter filament cleans of luminous matter a larger area around it, with radius $r \propto \lambda$. The baryonic matter so accreted forms ordinary galaxies, so a dynamical (not exact) baryonic Tully–Fisher law follows, $M_{\text{luminous}} \propto v^4$.

The force is proportional to the filament linear density and inversely proportional to the distance, $F \propto \frac{\lambda}{r}$, as per Equation (14). Therefore, the point at which the forces from two nearly parallel cylinders equilibrate is given by

$$\frac{r_1}{r_2} = \frac{\lambda_1}{\lambda_2} \, . \tag{35}$$

Thus, each dark filament cleans of gas an area extending out to $r \propto \lambda$. That area, taking a length $h$ along the $OZ$ axis, spans a volume $\pi r^2 h$; thus, it contains an amount of luminous matter given by

$$M_{\text{luminous}} = \pi r^2 h \bar{\rho}_{\text{luminous}} \, . \tag{36}$$

Thus, $M_{\text{luminous}} \propto \lambda^2$ and, invoking Equation (17), that makes $\lambda \propto v^2$,

$$M_{\text{luminous}} \propto v^4 \, . \tag{37}$$

Thus, the prediction of a cylindrically symmetric distribution of dark matter (or whatever gravitational source) coincides with that of Modified Newtonian Dynamics, that also yields an exponent of 4.

If we were to repeat the same reasoning for a spherical distribution, we would first have to argue for a density distribution $\rho \propto 1/r^2$ to yield a constant galactic rotation velocity (as in Equation (8)). A typical average density of dark matter $\bar{\rho}$ would follow from it, and $M \propto R^3$ would apply. Independently of that profile, taking $v_\infty$ beyond the end of the spherical distribution, $v_\infty \propto \sqrt{M}/\sqrt{R}$ (Kepler's third law), would then give $v_\infty \propto M^{1/3}$.

This proportionality would apply for all matter, luminous or otherwise. To convert it to a relation involving $M_{\text{luminous}}$, we would need to consider that the (larger) volume cleaned in the accretion of luminous matter to the dark lump would have $R_{\text{accretion}}^2 \propto M$ from the force law from Equation (1). Since $M_{\text{luminous}} \propto R_{\text{accretion}}^2$, we would conclude that $M_{\text{luminous}} \propto M^{3/2}$. Taking this to $v_\infty$ would finally yield

$$M_{\text{luminous}} \propto v^{9/2} \quad (\text{sphere}). \tag{38}$$

This is larger than and distinguishable from Equation (37). Obtaining other powers is possible with different assumptions, but the bottom line is that the sphere's relation has a larger exponent than the cylindrical one due to the different force law.

As typical data gives an exponent somewhere between 3 and 4 [27], it would perhaps suggest elongated geometry.

### 5.2. Scaling down to the Solar System

In this subsection, alone, I discuss a point of view that is disfavored by mounting evidence from the bullet-cluster and galaxies devoid of dark matter, that the extra gravitation might still be due to a phenomenon attaching to ordinary matter (such as strings coupling

to either of mass or its proxies, baryon or lepton number). This assumption elevates the Tully–Fisher relation from a dynamical effect to an actual law.

If that is the case, one could wonder how much would the effect be at solar system scales: could it be possible that a dark matter filament extended out of the solar poles and it had not been detected? This is actually not so easy to discard from planetary rotation measurements alone.

First, let us observe that the precision in the measurement of planet's distances $r$ ($a$), for example, Mercury, reaches $10^{-8}$ AU. Additionally, the measurement of their velocities is precise to $10^{-2}$ km/s. This yields a sensitivity floor on the $v(r)$ diagram below which no new effect would be visible yet (shaded band in Figure 8).

Because, under the assumption that the Tully–Fisher relation is exact and not a dynamical effect, $\lambda \propto M$, the known linear density of the filament associated to the entire galaxy can be rescaled to what would correspond to the solar system by means of $v \propto \sqrt{\lambda} \propto M$, so that, taking the ratio of the solar to the galactic mass,

$$v_{SS} = v_{\text{galaxy}} \sqrt{\frac{1 M_\odot}{(25.7 \pm 2.3) \times 10^{10} M_\odot}} \, . \tag{39}$$

Taking the resulting (dashed line) "flat velocity" to the solar system graph in Figure 8, we see that it comfortably lies two orders of magnitude below the precision achieved (which is already good enough to require isolation of much larger effects from conventional few-body classical mechanics).

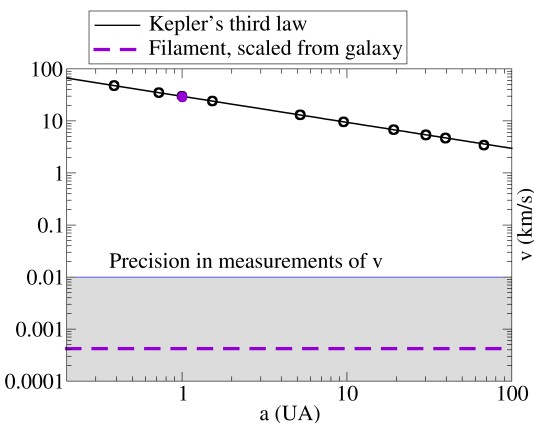

**Figure 8.** If the Tully–Fisher relation was an exact law and dark matter filaments were somehow attached to luminous matter, extrapolation of the flat rotation velocity (dashed line) from galaxy to solar mass would make the resulting velocity contribution too small to be detected: it lies well below Kepler's curve and the achieved precision in measuring planet velocities.

Of course, in conventional dark matter scenarios, there is no reason to believe any particular concentration of such material near the solar system, and no effect is expected here.

### 5.3. Inference by Stellar Scattering

In this subsection, I turn to the detectability in our own galaxy via surveys, such as GAIA [28,29]. The idea is that, should filamentary overdensities of dark matter exist, even if they are not directly visible, unlike filamentary gas structures [30], they might still be inferable by the behavior of nearby objects.

The theory for the scattering by a Newtonian $1/r^2$ force is well-known [31], and the geometry is depicted in Figure 9.

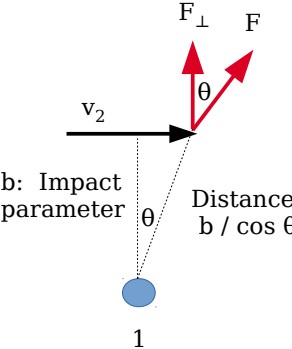

**Figure 9.** Geometry for the momentum transfer in a collision in the impact approximation [31] of a star (2) on either a spherical or a perpendicular cylindrical source (1), in the latter's reference frame. The trajectory along $v_2$ is practically straight and with constant velocity, with the transferred momentum approximately perpendicular to it.

Therein, the impact parameter is $b$, and the perpendicular force at a visual angle $\theta$ is

$$F_\perp = \cos\theta \frac{Gm_1m_2}{(b/\cos\theta)^2} . \tag{40}$$

If the scatterer is not a spherical body but, rather, a cylindrical one with linear mass density $\lambda_1$, the force law is modified to

$$F_\perp = \cos^2\theta \frac{2m_2\lambda_1 G}{b} . \tag{41}$$

The one less power of $1/r$ (see Equation (14)) entails here one less power of $\cos\theta$, that yields an inconsequential numeric factor but, more importantly, one less power of $b^{-1}$ in the denominator.

The momentum transfer to the projectile can be obtained by integrating the instantaneous impulse,

$$\Delta p_\perp = \int_{-\infty}^{+\infty} dt\, F_\perp ; \tag{42}$$

the integration time can be eliminated by the visual angle that is swiped along the (almost straight) scattered trajectory by

$$v_2 dt = d(b\tan\theta) . \tag{43}$$

Carrying the integration out with the $1/r^2$ or the $1/r$ forces just given yields

$$\Delta p_\perp = \frac{2Gm_1m_2}{bv_2} \quad \text{(spherical scatterer)}, \tag{44}$$

$$\Delta p_\perp = \frac{2\pi G\lambda_1 m_2}{v_2} \quad \text{(cylindrical scatterer)} . \tag{45}$$

The change from standard spherical scattering is the replacement of $m_1$ by $\pi\lambda b$. This entails that Equation (45) is independent of the impact parameter! That provides an interesting way of identifying such events: all stars in a small swarm or stellar stream change their velocity in the same amount (which is a feature that can be triggered on by Gaia's radial velocity measurements) and the ensemble of star's maintains its shape, as all its members slightly turn their velocities by the same amount.

As for the smallest filament so-detectable, solving for $\lambda$ and using $v_{2\perp} = \frac{p_{2\perp}}{m}$, we have

$$\lambda = \frac{v_2 \Delta v_{2\perp}}{2\pi G}; \tag{46}$$

the quantity $v_2 \Delta v_{2\perp}/\pi$ can be reasonably measured by GAIA down to 10 $(\mathrm{km/s})^2$ (given typical peculiar velocities of 30 km/s and a measurement precision of 1 km/s, that limits the accessible $\Delta v_{2\perp}$).

Since this directly yields the $(2G\lambda)$ factor of the scatterer, that can be compared to a galaxy's $(2G\lambda)$ of order $(250 \,\mathrm{km/s})^2$, we see that only scatterers by cylindrical overdensities of order $1\% - 0.1\%$ of the galactic ones will be conceivably detectable.

This discussion is relevant to distinguish whether the galactic dark matter halo, be it or not vertically elongated, can be composed of dark gas (WIMPS) or dark spherical bodies (MACHOS), for example, in which case this peculiar $b$-independent scattering will not be present, or whether there is a filamentary structure of dark matter at yet smaller scales than the galactic one. The peculiar motion of stars arises from random influences by stars and gas clouds (local overdensities above the average), so obviously it does not help with the global dark matter halo properties, but it would help with subhaloes or structures large enough, and in the context of this manuscript, with subfilaments or generally elongated overdensities.

A second handle to such potential substructure comes from the bound state problem instead of scattering. It falls off from the obvious observation that stars can orbit such subfilaments in an epicyclical fashion, with a velocity around the galactic center of order $\sqrt{2G\lambda}$ and a secondary velocity $\sqrt{2G\lambda'}$. Such bound stars perform a secondary oscillation that, crucially, is independent of the distance to the source. Thus, if such subfilaments exist, they are characterizable.

### 5.4. Direct Imaging

Galaxy catalogues, such as the Sloan Digital Sky Survey [32], clearly show that galaxies extend in filamentary structures forming a "cosmic web". At a scale of 40 Mpc, computer simulations do show a matching cosmic web of dark matter, where such linear structures are very prominent [33]. Zooming into smaller scales, it would appear that the filaments do extend, in the simulations, from galaxy to galaxy (see a beautiful illustration, last accessed on 10 September 2021, in https://skymaps.horizon-simulation.org/html/hz_AGN_lightcone.html).

It also appears that, looking back in time, one can also discern the filaments at a relatively smaller scale, directly linking galaxies, from observational data [34].

Further, though gravitational lensing has not been yet used to claim a filament at a galactic scale, a literature search reveals that a statistical stacking of galaxy pairs, with a rescaling performed so that all pairs sit on top of each others, shows evidence for dark matter filaments extending between neighboring galaxies [3], as mentioned above in Section 1. In addition to the imaged filament, these authors report that up to $1.5 \times 10^{13}$ Solar masses could be contained in such a filament, as per their lensing data. If this turns out to be statistically robust, it would eventually account for most of the dark matter, not leaving too much for spherical haloes. Further confirmation would entail that dark matter accretion has not evolved as much as usually assumed (from a homogeneous medium, to flat sheets, to filaments, to spherical structures: this last step would not be completed at today's $t_0$ cosmic time).

In the end, in spite of the many investigations addressing dark matter filaments, neither do those authors (nor many others) seem to have remarked the importance of the longitudinal structures that they were revealing for explaining galaxy rotation curves.

## 6. Further Consequences

### 6.1. Galaxy Plane—Galaxy Plane Correlations

If dark matter filaments extend between two or more galaxies, as revealed by both simulations and statistical stacking of galaxies in lensing studies, as discussed in Section 5.4, the two galaxy planes are correlated because both are preferentially perpendicular to the filament, as illustrated in Figure 10.

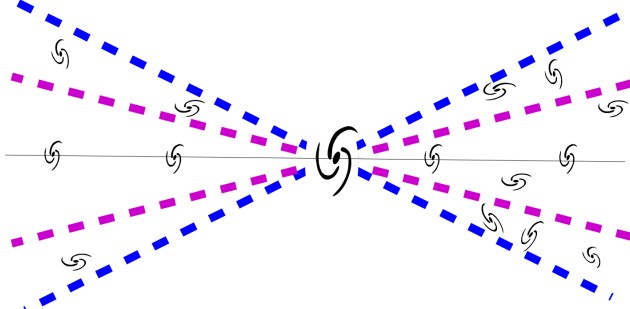

**Figure 10.** If a filament threads two or more galaxies before significantly bending, their rotation planes (perpendicular to the dark matter rope tying them) are parallel. Since the filaments are not visible in principle, one needs to correlate a distribution of galaxies. However, knowing that it is perpendicular to the spiral rotation plane, noise-reduction strategy is to not include all the space surrounding a given galaxy but, rather, to span only a cone given by a certain opening angle $\theta$ and then tighten that angle to improve the correlation.

As a measure of that correlation, one can take a sample of galaxies in a given volume and average the absolute value of their relative orientation cosine,

$$\xi = |\langle \hat{n}_1 \cdot \hat{n}_2 \rangle| . \tag{47}$$

In an infinite sample of randomly oriented galaxies, this number tends to zero. However, the approach to zero is slower if a few galaxies have oriented planes of rotation. To illustrate it, I have performed a simple calculation, shown in Figure 11. Spheres of increasing radius up to 10 Mpc are taken around a given galaxy, containing 1 galaxy/Mpc$^3$ with its plane randomly oriented. The blue squares show that, indeed, the average over all pairs of their relative orientation cosine quickly vanishes upon averaging over a sphere with the radius $r$ indicated in the $OX$ axis.

A line of a few equidistant, parallel galaxies, is then added over the north pole and under the south pole (red circles). Finally, the average is limited not to the whole sphere out to $r$, but to a cone of polar angle

$$\theta \in [0, \theta_{\max}] \cup [\pi - \theta_{\max}, \pi], \tag{48}$$

where the $\theta_{\max}$ varies between plots. The resulting correlation is significantly larger for small $r$.

The observable so-constructed is clear, but its interpretation is more ambiguous, as it can also be obtained by tidal effects of a different type. It has, indeed, been shown in simulations, such as Illustris-1 [35] or Horizon [36]. Even observational evidence has been claimed for a while now [37] and looks reminiscent of Figure 11.

*6.2. Virial Theorem for Small Galaxy Clusters*

A further comment that deserves attention is related to the virial, often used to extract the mass of galaxy clusters. If the time scale characteristic of the motion, $\tau$, allows for filament-filament interaction to have virialized, because the filaments are extended, their interaction is closer to a 2-dimensional gas instead of a three-dimensional gas of spheres. In that case, the standard

$$\langle T \rangle = \frac{-1}{2} \sum \mathbf{F}_k \mathbf{r}_k \tag{49}$$

that yields, for Newtonian potentials,

$$\langle T \rangle = \frac{-1}{2} \langle V \rangle \tag{50}$$

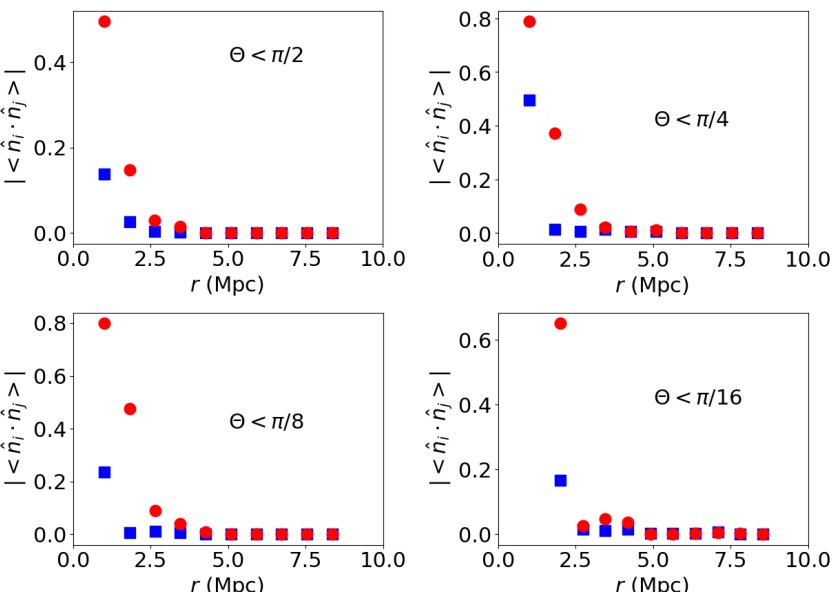

**Figure 11.** Toy simulations of a galaxy plane—galaxy plane correlation as function of the correlation distance. From top to bottom, the polar angle cut is tightened around the filament axis, so that the cone included features less random galaxies (but always the random ones along the filament). Blue squares: just the random galaxies. Red circles: the filament is now posited to thread perfectly oriented galaxies spaced at 1 Mpc, which are added to the sample from which the correlation is calculated.

is modified to its reduced-dimensional form with $F \propto 1/r \Rightarrow$,

$$\langle T \rangle \sim \text{constant}, \tag{51}$$

without a power dependence on the potential $V$. I have not studied whether $\tau$ is smaller enough than $1/H$ to have allowed such large structures to virialize.

*6.3. Gravitational Lensing*

The dark matter distribution of galactic haloes can be assessed with gravitational lensing, and, their shape is particularly amenable to study. The basic theory of lensing by an arbitrarily distorted mass distribution has recently been put forward by Turyshev [38], in terms of a spherical harmonic expansion. This allows studies of lensing with high distortion; in the extreme case of cylindrical distributions, one can use the geodesics of cylindrical solutions to Einstein's equations [39].

For smaller distortions, limited to an ellipticity in the dark matter halo of a localized population of galaxies, existing work has, indeed, extracted a significant deformation from Hubble space telescope data on gravitational lensing [40]. No specific analysis for spiral galaxies, where rotational curves have been measured, has been carried out.

This is an attractive venue for specialists in theoretical physics that will be revisited.

**7. Discussion**

Adopting the point of view that the source of the extra gravitational field driving galactic rotation curves is cylindrically distributed, or very elongated, their flatness is automatic (it is just Kepler's third law in one less dimension). No fine tuning of the dark matter distribution is needed; and the gravitational force remains the natural one in which the flux of the gravitational field is the same across concentric surfaces and the gravitational force just falls off because of its dilution (consistently with Gauss's law and Newtonian gravity). I believe this is a very educational exercise.

Whatever dark matter may fundamentally be, having it distributed with a spherical geometry is not a necessity. In fact, a cylindrically-symmetric distribution explains galactic

rotation curves just as well as a halo and allows for disposing of a hypothesis, such as temperature equilibration across that halo.

There is ample evidence for a filamentary structure at large scales in the universe, so the tenet is not in contradiction with most of the literature. The point of view, that those filaments are relevant at the kpc galactic scale, has not been observed in the literature, at least not widely enough to call for a solution of the galaxy rotation problem, though investigations of "fuzzy dark matter" [41] seem to be favoring more filamentary structures.

The nature of such hypothetical filaments or elongated haloes remains, such as generally that of dark matter, unknown. Much work has been devoted to cosmic strings and their networks [42]. Alternatively, more conventional dark matter, thought of as unspecified gravitating "stuff", could be accreting and, still today, be organized in longitudinal structure rather than more spherical haloes. In that case, these flat rotation curves are a temporary effect: as accretion continues from cylinders into spherical haloes, they will look more "Keplerian" in another few Gyr.

In any case, this fun work shows that galactic rotation curves are natural in the analytic limit in which the gravitational source is cylindrical. Thus, simulations of dark matter haloes would improve the rotational-data fit quality if the outcome distributions were much more prolate [43] than usually considered in this context. Fuzzy dark matter simulations may be a worthwhile endeavor.

**Funding:** Work supported with funding by grant MINECO: FPA2016-75654-C2-1-P (Spain); MICINN grants PID2019-108655GB-I00, -106080GB-C21; and Univ. Complutense de Madrid under research group 910309 and the IPARCOS institute.

**Conflicts of Interest:** The author declares no conflict of interest.

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
