# Peer review of "Elongated Gravity Sources as an Analytical Limit for Flat Galaxy Rotation Curves"

_universe, doi:10.3390/universe7090346_

Round 1

Reviewer 1 Report

See attached file

Author Response

See attached response

Reviewer 2 Report

Referee Report on the manuscript universe-1361434

This is my report on the Review " Elongated gravity sources
as an analytical limit for flat galaxy rotation curves", by Felipe J. Llanes-Estrada

The author reviews the main theoretical aspect behind the flatness of the rotation curve in spiral galaxies. Author argues that filamentary structures of the main mass component of the galaxy may explain in a very natural way the flatness of rotation curves. Nevertheless, filaments are not yet detected at small scales, and simulations do not have enough resolution to predict them.

I found the argument very interesting but the analysis is poor. Therefore, I cannot recommend the manuscript for publication in its current form.

=====================
  MAJOR CORRECTIONS
=====================

1) One concern I have is that the kinematics of stars in galaxies is the results of the existence and interaction of different component. How, for example, the kinematic of stars in the disk would be affected by the existence of such filamentary structures?

2) If those filaments exist, which is the correction factor expected in the proper motion of stars? Is it detectable in Milky Way with Gaia data?

3) Gravitational lensing strongly depends on the distribution of dark matter in galaxies. How would those filaments affect it?

Author Response

See attached response

Round 2

Reviewer 2 Report

Dear Editor,

the authors answered satisfactorily to my comments. Therefore, I recommend the manuscript for publication.